# MULTI-RESOLUTION MODELING OF A DISCRETE STOCHASTIC PROCESS IDENTIFIES CAUSES OF CANCER

**Adam Yaari**[*,1,2,3], **Maxwell Sherman**[*,1,3,5], **Oliver Priebe**[1,6],
**Po-Ru Loh**[3,5], **Boris Katz**[1,2], **Andrei Barbu**[1,2], **Bonnie Berger**[1,4]
[1] MIT CSAIL, [2] MIT CBMM, [3] Broad Institute of MIT and Harvard, [4] MIT Department of Mathematics,
[5] Division of Genetics, Brigham and Women's Hospital [6] Department of Physics, University of Pennsylvania
`{yaari,maxas,priebeo,boris,abarbu,bab}@csail.mit.edu`, `poruloh@broadinstitute.org`

## ABSTRACT

Detection of cancer-causing mutations within the vast and mostly unexplored human genome is a major challenge. Doing so requires modeling the background mutation rate, a highly non-stationary stochastic process, across regions of interest varying in size from one to millions of positions. Here, we present the split-Poisson-Gamma (SPG) distribution, an extension of the classical Poisson-Gamma formulation, to model a discrete stochastic process at multiple resolutions. We demonstrate that the probability model has a closed-form posterior, enabling efficient and accurate linear-time prediction over any length scale after the parameters of the model have been inferred a single time. We apply our framework to model mutation rates in tumors and show that model parameters can be accurately inferred from high-dimensional epigenetic data using a convolutional neural network, Gaussian process, and maximum-likelihood estimation. Our method is both more accurate and more efficient than existing models over a large range of length scales. We demonstrate the usefulness of multi-resolution modeling by detecting genomic elements that drive tumor emergence and are of vastly differing sizes.

## 1 INTRODUCTION

Numerous domains involve modeling highly non-stationary discrete-time and integer-valued stochastic processes where event counts vary dramatically over time or space. An important open problem of this nature in biology is understanding the stochastic process by which mutations arise across the genome. This is central to identifying mutations that drive cancer emergence (Lawrence et al., 2013).

Tumor drivers provide a cellular growth advantage to cells by altering the function of a genomic element such as a gene or regulatory feature (e.g. promoter). Drivers are identifiable because they reoccur across tumors, but there are two major challenges to detecting such recurrence. First, driver mutations are rare and their signal is hidden by the thousands of passenger mutations that passively and stochastically accumulate in tumors (Stratton et al., 2009; Martincorena & Campbell, 2015). Second, because functional elements vary dramatically in size (genes: $10^3$-$10^6$ bases; regulatory elements: $10^1$-$10^3$ bases; and single positions), driver mutations accumulate across regions that vary many orders of magnitude. Accurately predicting the stochastic accumulation of passenger mutations at multiple scales is necessary to reveal the subtle recurrence of driver mutations across the genome.

Here, we introduce the split-Poisson Gamma (SPG) process, an extension of the Poisson-Gamma distribution, to efficiently model a non-stationary discrete stochastic process at numerous length scales. The model first approximates quasi-stationary regional rate parameters within small windows; it then projects these estimates to arbitrary regions in linear time (10-15 minutes for genome-wide inference). This approach is in contrast to existing efforts that model fixed regions and require computationally expensive retraining (e.g. over 5 hours) to predict over multiple scales of interest (Nik-Zainal et al., 2016; Martincorena et al., 2017). We apply our framework to model cancer-specific mutation patterns (fig. 1). We perform data-driven training of our model's parameters and show that it more accurately captures mutation patterns than existing methods on simulated and real data. We demonstrate the power of our multi-resolution approach by identifying drivers across functional

---

[*]Authors contributed equally to this work.

elements: genes, regulatory features, and single base mutations. Despite the method having no knowledge of genome structure, it detects nearly all gene drivers present in over 5% of samples while making no false discoveries and detects all previously characterized regulatory drivers. Detected events also include novel candidate drivers, providing promising targets for future investigation.

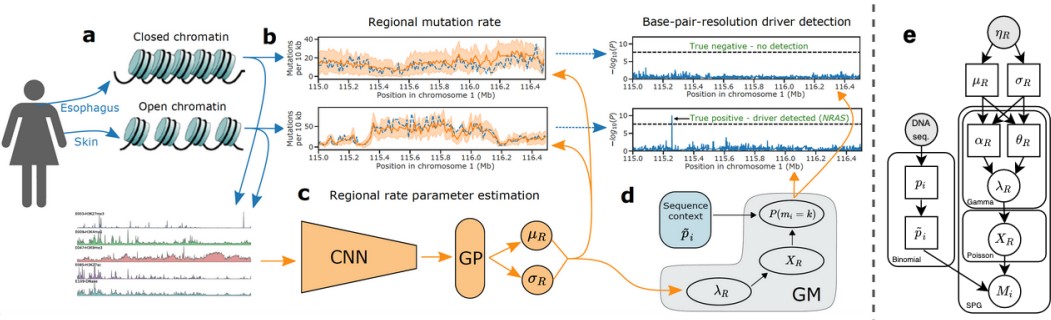

Figure 1: Non-stationary stochastic process modeling predicts mutation patterns and identifies cancer-specific driver mutations. Biological processes are shown in blue, data processing is shown in orange. **a.** Areas of the genome have varying epigenetic states (e.g. accessibility for transcription) depending on the tissue type. **b.** These epigenetic states set different mutation rates in different tissues. **c.** Our model takes these epigenetic tracks as input to estimate the regional mutation density across the genome (95% confidence interval in orange). **d.** Regional rate parameters and sequence context are integrated via the split-Poisson-Gamma (SPG) distribution to provide arbitrary resolution mutation count estimates. Deviations between the estimated and observed mutation rates identify mutations that are associated with cancers in different tissues. **e.** The split-Poisson-Gamma (SPG) model plate diagram (squares: inferred parameters; grey: observed input data).

## 1.1 PREVIOUS WORK

Numerous methods exist for modeling stationary stochastic processes (Lindsey, 2004). Far fewer exist for non-stationary processes because they are difficult to capture with the covariance functions of parametric models (Risser, 2016). Non-stationary kernels have been introduced for Gaussian processes (Paciorek & Schervish, 2004), but these may not be tractable on large datasets due to their computational complexity. More recently, there has been work developing Poisson-gamma models for dynamical systems (Schein et al., 2016; Guo et al., 2018), but these methods have focused on learning relationships between count variables, not predicting counts based on continuous covariates.

In the particular case of modeling mutation patterns across the cancer genome, numerous computational methods exist to model mutation rates within well-understood genomic contexts such as genes (Lawrence et al., 2013; Martincorena et al., 2017; Wadi et al., 2017; Mularoni et al., 2016; Juul et al.). These models account for $< 4\%$ of the genome (Rheinbay et al., 2020). They are not applicable in non-coding regions, where the majority of mutations occur (Gloss & Dinger, 2018). A handful of methods to model genome-wide mutation rates have been introduced (Polak et al., 2015; Nik-Zainal et al., 2016; Bertl et al., 2018). However, they operate on a single length-scale or set of regions and require computationally expensive retraining to predict over each new length-scale. Several methods rely on Poisson or binomial regression; however, previous work has extensively documented that mutation counts data are over-dispersed, leading these models to underestimate variance and yield numerous false-positive driver predictions (Lochovsky et al., 2015; Martincorena et al., 2017; Juul et al., 2019). Negative binomial regression has recently been used to account for over-dispersion (Nik-Zainal et al., 2016) and perform genome-wide mutation modeling and driver detection. However, resolution was coarse, and it only found a few, highly recurrent driver mutations.

## 1.2 OUR CONTRIBUTIONS

This work makes three key contributions: 1) we introduce an extension of the Poisson-Gamma distribution to model non-stationary discrete stochastic processes at any arbitrary length scale without retraining; 2) we apply the framework to capture cancer-specific mutation rates with unprecedented accuracy, resolution, and efficiency; and 3) we perform a multi-scale search for cancer driver mutations genome-wide, including the first-ever base-resolution scan of the whole genome. This search yields

several new candidate driver events in the largely unexplored non-coding genome, which we are working on validating with experimental collaborators. Crucially, our approach allows fast, efficient, and accurate searches for driver elements and mutations anywhere in the genome without requiring arduous retraining of a model, a feat which is not possible with existing approaches.

## 2 MULTI-RESOLUTION MODELING OF A NON-STATIONARY DISCRETE STOCHASTIC PROCESS

We consider a non-stationary discrete stochastic process $\{M_i; i = 1, 2, ...\}$ where $M_i$ is the integer-valued event count at position $i$. Associated with each position $i$ is a real-valued, L-dimensional feature vector $\eta_i$ that determines the instantaneous event rate $\lambda_i$ via an unknown function. Thus a region $R = \{i, i+1, ..., i+N\}$ of $N$ contiguous positions is characterized by an $L \times N$ feature matrix $\eta_R$ and an event count $X_R = \sum_{i \in R} M_i$. As training data, $\eta_R$, $X_R$, and $M_i$ are observed for some set of regions $\{R \in \mathscr{T}\}$. Then given a set of feature matrices from unobserved regions $\{\eta_R; R \in \mathscr{H}\}$, the challenge is to predict the distribution of event counts over any arbitrary set $I$ of unseen positions that may or may not be contiguous. Real-world examples include traders in a stock market, packets delivered to routers in a network, and mutations accumulating at positions in the genome.

### 2.1 THE SPLIT-POISSON-GAMMA PROCESS

We assume that the process is near-stationary within a small enough region $R = \{i, i+1, ..., i+N\}$ and that the $L \times N$ covariate matrix $\eta_R$ is observed. Thus the rate of events $\lambda_R$ within $R$ is approximately constant and associated with $\eta_R$, albeit in an unknown way. A number of events ($X_R$) may occur within $R$ dependent on $\lambda_R$ and are then stochastically distributed to individual positions within $R$, implying a hierarchical factorization of the scalar random variables $\lambda_R$, $X_R$, and $M_i$ (fig. 1e) as

$$Pr(M_i = k, X_R, \lambda_R; \eta_R) = Pr(M_i = k | X_R; \eta_R) Pr(X_R | \lambda_R; \eta_R) Pr(\lambda_R; \eta_R). \tag{1}$$

$X_R$ and $\lambda_R$ are unknown nuisance variables and are marginalized in general as

$$Pr(M_i = k | \eta_R) = \int_0^\infty Pr(\lambda_R; \eta_R) \sum_{X_R = k}^\infty Pr(M_i = k | X_R; \eta_R) Pr(X_R | \lambda_R; \eta_R) d\lambda_R. \tag{2}$$

Since applications often require many posterior predictions over regions of varying sizes, we propose a prior parameterization that builds on the success and flexibility of the classical Poisson-Gamma distribution while ensuring the marginalization has an easy-to-compute posterior distribution:

$$\lambda_R \sim \text{Gamma}(\alpha_R, \theta_R) \tag{3}$$
$$X_R \sim \text{Poisson}(\lambda_R) \tag{4}$$
$$M_i \sim \text{Binomial}(X_R, \tilde{p}_i) \tag{5}$$

where $\alpha_R$ and $\theta_R$ are shape and scale parameters dependent on $\eta_R$, $p_i$ is the time-averaged probability of an event at $i$ and $\tilde{p}_i = \frac{p_i}{\sum_{j \in R} p_j}$, the normalized probability within $R$. A plate diagram of the hierarchical model is presented in fig. 1e.

The above formulation provides a simple, closed form solution to eq. (2) as a negative binomial (NB) distribution (See Appendix for details):

$$Pr(M_i = k | \alpha_R, \theta_R, \tilde{p}_i; \eta_r) = NB\left(k; \alpha_R, \frac{1}{1 + \theta_R \cdot \tilde{p}_i}\right). \tag{6}$$

Eq. 5 implicitly assumes that events are distributed independently to units within $R$. Exploiting this assumption, eq. (6) immediately generalizes to consider *any* set of units $I \subseteq R$ as

$$Pr\left(\sum_{i \in I} M_i = k | \alpha_R, \theta_R, \{\tilde{p}_i\}_{i \in I}; \eta_R\right) = NB\left(k; \alpha_R, \frac{1}{1 + \theta_R \cdot \sum_{i \in I} \tilde{p}_i}\right). \tag{7}$$

The above formulation is an extension of the classical Poisson-Gamma distribution whereby the Poisson is randomly split by a binomial. We term this a split-Poisson-Gamma (SPG) process. While

the derivation of the SPG solution makes simplifying assumptions, the benefit is that the parameters $\alpha_R$ and $\theta_R$ need to be estimated only once for each non-overlapping region $R$. *Estimates for a region of any other size can then be computed in constant time* from eq. (7). If a new region $R'$ is larger than $R$, we approximate the gamma distribution in a super-region containing $R'$ as a superposition of the previously inferred parameters of each region of size $R$ within the super-region (see section 2.2).

## 2.2 INFERRING REGIONAL RATE PARAMETERS

The statistical power of SPG depends on accurate estimation of the regional gamma rate parameters $\alpha_R$ and $\theta_R$. We propose a variational approach to enable flexible, accurate, and non-linear inference of these parameters from a set of covariates. Let $G(\alpha, \theta)$ be a gamma distribution. By the central limit theorem, $\lim_{\alpha \to \infty} G(\alpha, \theta) = N(\mu, \sigma^2)$ where $\mu = \alpha\theta$ and $\sigma^2 = \alpha\theta^2$. We thus use a Gaussian process (GP) to non-linearly map covariates to regional estimates for $\mu_R$ and $\sigma_R^2$. The variational estimates for the gamma parameters are then

$$\alpha_R = \mu_R^2/\sigma_R^2, \qquad \theta_R = \mu_R/\sigma_R^2 \qquad (8)$$

For a super-region $R' = R_i + R_j$, $\mu_{R'} = \mu_{R_i} + \mu_{R_j}$ and $\sigma_{R'}^2 = \sigma_{R_i}^2 + \sigma_{R_j}^2$.

A limitation of this approach is that GPs can only operate on vectors of covariates. Thus a dimensionality reduction method must be applied to the input matrix $\eta_R$. In cases where $\eta_R$ includes spatial relationships, a convolutional neural network can be a powerful approach to dimension-reduction; however, other approaches are feasible (see section 3.2 and section 5.1).

## 2.3 INFERRING TIME-AVERAGED EVENT PROBABILITIES

The time-averaged parameters $\{p_i; i = 1, 2, ...\}$ must also be inferred. Crucially, as seen in eq. (5), these parameters are never used directly; instead, they are always renormalized to sum to one within a region of interest. Thus, estimates do not need to reflect the absolute probability of an event at $i$ but merely the *relative* rate of events between positions. Indeed, because of the renormalization procedure, the estimates need not even be a true probability distribution. Estimating $p_i$ can thus be accomplished by clustering units with similar relative rates of events. How this clustering should be performed will depend on the application of interest (see section 3.3 for a concrete example).

## 3 FITTING PARAMETERS TO PREDICT CANCER MUTATION PATTERNS

We obtained publicly available mutation counts from four cancer cohorts previously characterized by the Pan-Cancer Analysis of Whole Genomes Consortium (PCAWG) (Campbell et al., 2020): esophageal adenocarcinoma (N = 98 tumors; n ≈ 2.7M mutations), skin melanoma (N = 70 tumors; n ≈ 7.8M mutations), stomach adenocarcinoma (N = 37 tumors; n ≈ 480k mutations), and liver hepatocellular carcinoma (N = 264 tumors; n ≈ 3.3M mutations). Crucially, these data contain only the total number of mutations at each position in the genome. *We do not know a priori which mutations are background mutations and which are driver mutations. We also do not know the true mean and variance of the underlying mutation rate in any region.*

We do know that the mutation rate is highly associated with chemical modifications of the DNA that set the way it is processed in a cell, collectively termed the epigenome (Schuster-Böckler & Lehner, 2012; Polak et al., 2015). We obtained 733 datasets characterizing the patterns of these chemical modifications in 111 human tissues from Roadmap Epigenomics (Roadmap Epigenomics Consortium et al., 2015). These data are the largest compendium of uniformly processed human epigenome sequencing currently available. Each track provides the -$\log_{10}$ P-value that a particular modification is present at each location of the genome in a given tissue type. We additionally created two tracks that provide the average nucleotide and GC content in a region based on the human reference genome GRCh37. See Appendix and supplementary data for additional information on the epigenetic tracks. The input matrix for each region $\eta_R$ thus has 735 rows. We fixed the number of columns to be 100 irrespective of the size of $R$, where each column is the mean across $R/100$ adjacent positions.

### 3.1 ARTIFICIAL DATASET

In order to evaluate the ability of SPG and other models to estimate the unknown mean and variance of regional rates, we created simulated datasets with known mean and variance parameters dependent

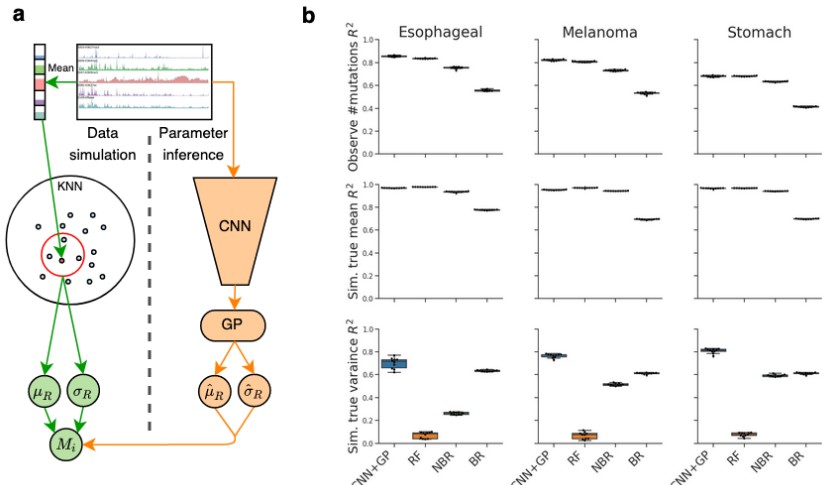

Figure 2: Data simulation and regional parameters inference accuracy across methods. **a.** Simulated data experiment. Data simulation is shown in green, parameter inference is shown in orange. Simulated $\mu_R$ and $\sigma_R$ are computed as the mean and variance of a sample's KNN cluster. The model is trained over randomly sampled event counts; the parameter estimates $\hat{\mu}_R$ and $\hat{\sigma}_R$ are then compared to their true values. **b.** Pearson $R^2$ to the observed mutation count in the true (**unsimulated**) data (top) simulated mean (middle) and simulated variance (bottom) for the CNN+GP parameter estimation strategy (CNN+GP), random forest (RF), negative binomial regression (NBR) and binomial regression (BR). Results for additional estimation techniques are in Appendix.

on the observed input matrix (fig. 2a). We created input matrices of size $735 \times 100$ from the epigenetic tracks (described above) for non-overlapping regions of 50,000 positions. To define non-stationary mean and variance of mutation rate dependent on the each region's input matrix, we reduced $\eta_R$ to an feature vector of size 735 by taking the mean across columns and used a k-nearest-neighbors (KNN) strategy to identify 500 regions with similar epigenetic feature vectors; we then defined $\mu_R$ and $\sigma_R^2$ for each region as the mean and variance of the observed event counts across its 500 neighbouring regions. The number of observed events for that region was then randomly drawn from a negative binomial distribution defined by those parameters (full technical details in Appendix). Models were trained on the randomly drawn counts and evaluated on their ability to accurately infer the true mean and variance. We simulated 50kb regions following previous work (Rheinbay et al., 2020).

## 3.2 ESTIMATING DYNAMIC REGIONAL RATES WITH UNCERTAINTY

The input matrices $\eta_R \in \mathbb{R}^{735 \times 100}$ required significant dimension reduction before we could employ our GP-based variational strategy to infer SPG regional rate parameters. Columns encode the high-resolution spatial organization of the epigenome which have recently been shown to be important determinants of local mutation rate (Gonzalez-Perez et al., 2019; Akdemir et al., 2020). Therefore, we hypothesized that a convolutional neural network (CNN) would provide a powerful approach to produce a low-dimensional embedding that retrains information about this local structure; the supervised nature of a CNN further enables the resulting embedding to be optimized for the cancer of interest, which is crucial to performance since the epigenetic determinants of mutation rate vary drastically between cancer types (Polak et al., 2015). We constructed a 1D CNN model with 4 residual blocks and 3 fully-connected layers to map mutation-rate-associated local epigenetic patterns to regional mutation rates. The CNN non-linearly reduces $\eta_R \in \mathbb{R}^{735 \times 100}$ to a 16 dimensional feature vector in its last feature layer. The CNN was trained to minimize the mean squared error between observed and predicted mutation counts. Due to the interchangeable nature of the rows, the 1D kernels allow the network to identify arbitrary inter-track interactions. The final 16-dimension feature vector was then passed as input to a sparse GP (Titsias), fit to maximize the likelihood of the observed mutation counts using 2000 inducing points and a radial basis function kernel (fig. 1b). We found that results were robust to the particular choice of kernel and hyperpriors placed over kernel parameters. While end-to-end training is possible (Bradshaw et al., 2017), we did not find it necessary to achieve high accuracy in this particular application. A CNN is not the only method available to reduce dimensionality prior to GP inference; we investigated numerous other methods, but found the CNN+GP to produce the most accurate results (see Appendix).

### 3.3 ESTIMATING TIME-AVERAGED EVENT PROBABILITIES

In the case of cancer mutation patterns, previous work showed that the mutation rate at any position $i$ is heavily influenced by the nucleotide at $i$ and the two nucleotides directly adjacent to $i$; positions with this same "trinucleotide context" will have similar mutation patterns (Alexandrov et al., 2013). Following previous works (Mularoni et al., 2016; Wadi et al., 2017; Martincorena et al., 2017; Weghorn & Sunyaev, 2017), we used trinucleotide context to estimate $p_i$. Let $ntn'$ be the trinucleotide context centered at position $i$. We estimate the probability that $i$ is mutated using the ensemble maximum-likelihood estimate of its cluster

$$p_i = p_{n,t,n'} = \frac{v_{n,t,n'}}{N_{n,t,n'}}. \tag{9}$$

where $N_{n,t,n'}$ is the number of $ntn'$ trinucleotides in the genome and $v_{n,t,n'}$ is the number of times $t$ is mutated within $ntn'$. This approach alone explains little variance in sub-megabase regions (see Appendix) because it does not account for regional mutation rates.

#### 3.3.1 COMPARING TO BENCHMARK MODELS

We compared SPG to three alternative approaches that have been previously used to learn both the mean and variance of regional mutation patterns genome-wide. The alternative models are random forest (RF) regression (Polak et al., 2015), binomial regression (BR) (Bertl et al., 2018), and negative binomial regression (NBR) (Nik-Zainal et al., 2016; Martincorena et al., 2017). For the RF, we used the Jackknife method (Wager et al.) to estimate the variance; this method requires $O(n)$ trees where $n$ is the number of samples in the training set. BR and NBR directly specify the variance as function of the mean: BR as $\sigma_R^2 = \mu - \mu^2/n$ and NBR as $\sigma_R^2 = \mu_R(1 + \beta\mu_R)$, where $\beta$ is an overdispersion parameter. Benchmarking comparisons were performed on the skin melanoma, esophageal adenocarcinoma, and stomach adenocarcinoma cohorts.

### 3.4 MODEL TRAINING

For every region of size $R$, epigenetic features were extracted into matrices of size 735 tracks by 100 binned position columns, where each column was the mean across $R/100$ adjacent base-pairs. Regions with highly repetitive DNA sequence (<70% of 36mer sub-sequences being unique) where excluded from the training set to ensure high data quality as in previous analyses (Polak et al., 2015). Before training, high-quality data regions were strictly split into train (64%), validation (16%) and test (20%) sets. Predictions for excluded regions and held-out test sets were obtained after model training. Genome-wide predictions were generated using 5-fold cross-validation. The CNN received the full $735 \times 100$ matrices as input. Vector-based methods (RF, NBR, BR) received the 735-dimension vector of epigenetic values averaged across position columns. Following previous work, we also included the expected number of mutations based on the trinucleotide composition of a region as an offset term in NBR and BR when predicting mutation counts (Nik-Zainal et al., 2016). Additional details on training (e.g. number of epochs) are in Appendix.

## 4 IDENTIFYING GENETIC DRIVERS OF CANCER

Because cancer drivers reoccur across tumors, driver elements (genes, regulatory structures, and individual base-pairs) will contain an excess of mutations relative to the of expected background mutations. The SPG model provides a simple, efficient, and accurate method to search for this recurrence. We first estimate mean and variance of the background mutation rate using the CNN+GP estimation method. We then apply eq. (7) to search for statistical evidence that the number of observed mutations, $k$, exceeds expectation within every gene, known regulatory structure, and 50 bp window in the genome by changing the set of tested positions $I$. For a gene, $k$ is the number of observed missense or nonsense mutations and $I$ is the set of all possible mutations in the gene. For both a regulatory element and window of fixed size, $k$ is the number of mutations observed in the element / window and $I$ is the set of all positions within the element / window. If an element overlaps multiple 10kb regions, we merge the mean and variance estimates for overlapped regions as described in section 2. To maintain strict train-test separation, both the rate parameters and $p_i$ are estimated excluding the element being tested. We controlled family-wise error rate at the $\alpha = 0.05$ level using a Bonferroni correction for the total number of tests in genes, regulatory elements, or 50bp windows.

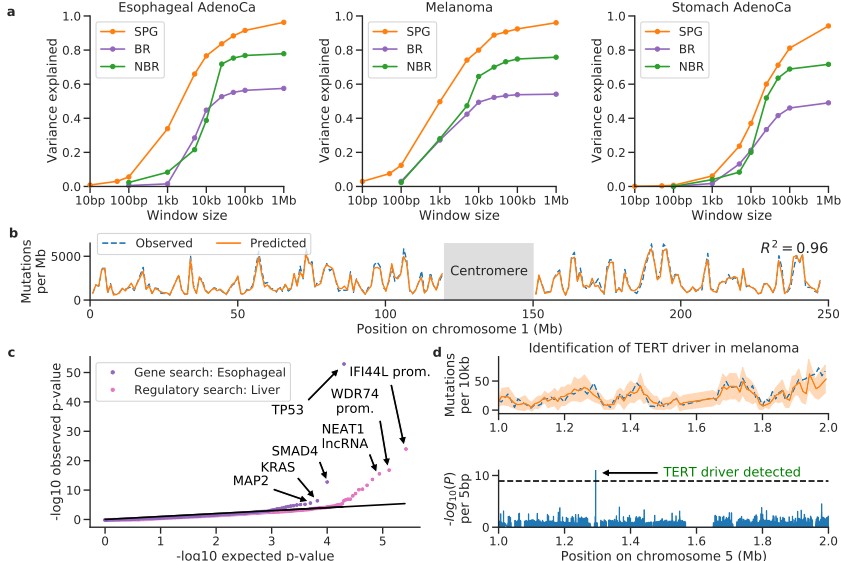

Figure 3: SPG accurately models mutation density and detects driver events. **a.** Variance explained (Pearson $R^2$) of the observed mutation count by our SPG, binomial regression (BR) and negative binomial regression (NBR) across length-scales. **b.** Observed (dashed blue) and predicted (solid orange) mutation density from our GM at 1Mb regions across chromosome 1 in melanoma. (**c**) Quantile-quantile plots of expected and observed P-values for gene driver detection in esophageal adenocarcinoma (purple) and driver regulatory element detection in liver cancer (pink). Esophageal and liver were chosen only for the sake of readability; qq-plots are similar for all cancers. (**d**) Model detection of a well-known non-coding driver in the *TERT* promoter in melanoma at 1kb resolution. Black dashed lines: Bonferroni-corrected genome-wide significance thresholds.

Gene information was obtained from Martincorena et al. (2017) and regulatory element information from Rheinbay et al. (2020). Driver detection was performed in all four cancer cohorts.

## 5 RESULTS

### 5.1 ACCURACY OF REGIONAL RATE PARAMETER ESTIMATION

We first evaluated various methods' abilities to infer regional rate parameters, considering both new (CNN+GP) and existing (RF, NBR, BR) methods. We assessed each method's ability to learn the expected mutation rate by directly assessing the amount of variance (Pearson $R^2$) it explained over observed mutation counts in 50kb real data windows (fig. 2b top), and found the CNN+GP estimation method performed the best, although random forest was a close second (results were similar when estimating the mean in simulated data; fig. 2b middle). We then evaluated each method's ability to capture the variance $\sigma_R^2$ in the simulated data, quantified as the Pearson $R^2$ to the true variance. The CNN+GP method again outperformed the others (fig. 2b bottom). Notably, RF was unable to infer the variance beyond chance level, and thus we did not consider this method further because its inability to infer variance precludes accurate driver detection.

We also considered other dimensionality reduction techniques including both non-neural and neural approaches as well as supervised and unsupervised approaches, as an alternative to the CNN; no other approach achieved accuracy comparable to the CNN+GP over both mean and variance (see Appendix). Moreover, we validated the necessity of the GP by directly optimizing the CNN to predict both parameters and found it significantly reduced model performance (7% decrease over mutation counts and 13% over $\sigma_R^2$ within 10kb windows in melanoma).

### 5.2 ACCURACY AND EFFICIENCY OF MUTATION RATE PREDICTION

To further compare the SPG performance to existing methods, we evaluated the accuracy and efficiency of each method over length scales ranging over 5 orders of magnitude ($10$-$10^6$ positions). To evaluate SPG, we estimated the background mutation rate parameters, $\mu_R$ and $\sigma_R^2$, in 10kb regions genome-wide using the CNN+GP estimation strategy; we then applied the SPG distribution to estimate mutation count distributions over all other region sizes. The existing methods with

| Method | 100bp | 1kb | 10kb | 100kb | 1Mb | Multi-Scale |
|---:|---:|---:|---:|---:|---:|---:|
| SPG | 4m11s | 3m33s | 36m35s | 19s | 2s | **42m40s** |
| NBR | 1h30m | 7m3s | 43s | 6s | 4s | 1h37m56s |
| BR | 44m36s | 3m8s | 15s | 5s | 4s | 48m8s |
| RF | >15h | >15h | 14h2m | 5m24s | 28s | >15h |

Table 1: Run times for SPG, NBR, BR, and RF for five region sizes and multi-scale search. Reported times are for a single train-validation-test split per model over 8 CPUs and 1 GPU machine. For SPG, parameters were inferred using the CNN+GP estimation method at 10kb, running the CNN and GP one time each; hence SPG's increased run time at 10kb relative to other region sizes. Bolded is the best multi-scale search time. Presented RF times are high due to the need for O(n) trees to estimate variance using the Jackknife method.

reasonable performance on both mean and variance prediction (BR and NBR) were trained to directly predict the count distribution in each region for each length scale genome-wide.

Across all tested window sizes and cancers, SPG outperformed existing methods, with performance particularly improved in esophageal adenocarcinoma and skin melanoma (fig. 3a), crucial for high-accuracy driver detection downstream. Across 1Mb windows, SPG explains $> 95\%$ of the variance in mutation density across all three cancers (fig. 3a,b); this is >15% more variance than both existing methods (Fig. 3a), highlighting the ability of SPG to accurately capture regional distribution parameters and project them upwards. The decrease in variance explained in smaller window sizes is expected because observed mutation counts become increasingly stochastic relative to the expected number of mutations predicted by each method. The theoretical foundations of negative binomial regression and SPG are similar, both built upon the classical Poisson-gamma model. SPG differs from NBR in three key ways that help explain its improved performance: 1) SPG models mutation patterns over arbitrary sets of positions enabling it to dynamically pool information across positions after a single training; in contrast, NBR operates on fixed regions and must be retrained for every new region size. 2) SPG's variational inference method estimates the gamma parameters for each region independently; NBR estimates only the shape parameter independently for each window and uses a single scale parameter for all windows. 3) SPG's CNN data reduction enables non-linear mapping of spatial covariate information to mutation rate, whereas NBR can perform only linear inference and disregards the spatial organization of the genome.

SPG is also the most efficient method for multi-resolution search (appendix D.3). Initial training of parameters using the CNN+GP method for one fold of 10kb regions required 36 minutes using 1 GPU. Projection to each additional scale using 8 CPUs required at most 4 minutes (table 1). In contrast, training time for BR and NBR increases considerably as the resolution decreases. Performing a search across resolutions of 50bp, 100bp, 500bp, 1kb, and 10kb would require >5h for negative binomial, >2h for BR, and only 52 minutes for SPG (Appendix). We have also found that parameter estimation on windows as large as 100kb does not significantly reduce accuracy across scale (Appendix), allowing SPG parameter estimation in considerably shorter time (e.g. only 8 minutes for 50kb).

### 5.3 IDENTIFICATION OF CANCER DRIVER MUTATIONS

We leveraged SPG's ability to model multiple resolutions to search the whole genome of each of the four cancer cohorts for gene drivers, non-coding regulatory drivers, and 50bp windows that may harbor a driver mutation. All significant results are provided as supplementary data tables. We compared our results to those obtained from a previous comprehensive characterization of these cohorts by Campbell et al. (2020), who used 13 different methods to identify drivers. Our model did not have access to information about gene structure or function unlike the methods used in the previous characterization. Nonetheless, the model's p-values were well calibrated (fig. 3c), and we identified 19 genes with a significant excess of missense or nonsense mutations. All 19 genes were previously reported as drivers by Campbell et al. (2020). We failed to detect only two known driver genes present in >5% of samples. This performance is on par with state-of-the-art methods specifically designed for driver gene identification (Rheinbay et al., 2020).

When analyzing non-coding regulatory elements, SPG's p-values were again well calibrated (fig. 3c), and it identified all non-coding drivers (n=11) identified by Campbell et al. (2020). Moreover, SPG implicated several additional putative non-coding driver elements that had not been previously reported. Examples include 1) the promoter of the gene *MTERFD1* in esophageal cancer (P = $3.1 \times 10^{-8}$), whose over-expression has been observed in numerous cancers, has been shown to promote cell growth, and decrease clinical survival (Zhang et al., 2014); 2) an enhancer of *DHX33* in

liver cancer ($P = 4.8 \times 10^{-11}$), whose over-expression has been shown to promote cancer development (Wang et al., 2019); and 3) the 5' UTR of *ERN1* in melanoma, which has been linked to cancer therapy resistance (Šuštić et al., 2018).

Finally, we performed the first, to our knowledge, genome-wide search for individual driver mutations. All significant genic hits fell within known driver genes whose functions have been experimentally validated including *TPF3*, *BRAF*, *KRAS*, *PIK3CA*, and *CTNNB1*. In addition, SPG identified two recurrent mutations in the genes *GPR98* and *KLB* that had not been previously identified in Campbell et al. (2020)'s analysis of the data . These mutations are listed as driver mutations in the Catalogue of Somatic Mutations in Cancer Tate et al. (2018). SPG implicated numerous hotspots in the mostly unexplored non-coding genome, including the well-known *TERT* promoter mutation (fig. 3d). These results are promising targets for future studies of non-coding drivers in cancer cell lines and organoids.

## 6 DISCUSSION

We introduced an extension of the Poisson-Gamma distribution to model discrete-time, integer-valued stochastic processes at multiple scales. The split-Poisson-Gamma (SPG) model makes several simplifying assumptions including: 1) that the process is quasi-stationary in a small enough region; 2) events are distributed among the discrete units approximately independently; and 3) the behavior of the random variables can be captured by particular parametric distributions. The assumptions are necessary to derive a closed-form posterior distribution. This enables efficient prediction over multiple length-scales without having to re-estimate the model parameters. We additionally proposed a variational inference strategy to reduce input dimensionality and estimate the parameters of the model using a CNN coupled with a GP. Indeed, the use of a CNN+GP for a distribution variational inference may be of use well beyond the SPG framework and discrete stochastic process modeling.

To demonstrate the utility of the SPG, we applied it to model mutation rates in cancer and identify genomic elements that drive tumor emergence. In the case of this application, previous work has established the validity of the above assumptions, demonstrating that the mutation rate is approximately constant within 50kb regions (Rheinbay et al., 2020) and that mutations occur approximately independently given each position's trinucleotide context (Martincorena et al., 2017). We demonstrated that the approach is more accurate than other methods on both real and synthetic data. We also demonstrated that multi-resolution prediction enables identification of both known and novel putative drivers of cancer, including in the non-coding genome, a crucial open problem in genomics (Khurana et al., 2016; Rheinbay et al., 2020).

SPG is also applicable to discrete stochastic challenges in other domains, particularly when anomaly detection is the goal. For example, cybersecurity is often interested in detecting malicious network activity that may occur over seconds, hours, or weeks. Such activity ought to appear as anomalous relative to the expected network traffic. However, similarly to cancer drivers, detecting such anomalies is confounded by the fact that expected network traffic can vary dramatically over time. Thus detecting malicious activity requires modeling non-stationary event rates and searching for anomalous activity across multiple resolutions. SPG is highly suited for efficient execution of this task. While the details of parameter estimation will depend upon the application, we expect the variational Gaussian process approach will be broadly applicable and that, in the case of high-dimensional matrix input, a CNN will provide a powerful tool to reduce the data to an informative feature vector.

Another timely use-case is identifying infectious disease outbreaks. The task is to determine when a new infection hotspot is developing to implement containment measures. This task is challenging because infection rates vary by geography and by individuals (e.g. young vs. elderly). SPG provides a framework to identify infection hotspots while accounting for geographic and demographic risk. In this application, regional rate parameters would reflect geographic infection rates while individual people would be the "positions" of the stochastic process. The task would be to identify groups of people (e.g. schools, neighborhoods, cities, etc.) with more cases than expected. Local information (e.g. medical treatment availability, number of cases, population density, etc.) could serve as predictors to infer regional infection rates while an individual's demographics could provide the clustering criteria to estimate $p_i$. Such an application could be particularly useful for early identification of hotspot outbreaks of COVID-19. These examples highlight the diverse situations to which SPG could be applied, and we expect that the number of applications will continue to grow as collections of time- and space-varying data grow increasingly large.

ACKNOWLEDGEMENTS

We acknowledge the contributions of the many clinical networks across ICGC and TCGA who provided samples and data to the PCAWG Consortium and the contributions of the Technical Working Group for collation, realignment, and harmonized variant calling of the cancer genomes used in this study. We thank the patients and their families for their participation in the individual ICGC and TCGA projects.

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

## A  APPENDIX

In this appendix, we provide detailed information on:

1. The data used in this work including its origin and all preprocessing steps.
2. Additional method details including:
   - A derivation of the closed-form marginal distribution of the graphical model presented in main text.
   - Architecture and training details of all models.
   - How the genome-wide search for driver mutations was performed.

The appendix also includes an analysis of the sensitivity of negative binomial regression to detect well-known drivers genome-wide and additional figures that provide context to results presented in the main paper.

## B  DATA

### B.1  EPIGENETIC TRACKS

We obtained 733 $-log_{10}$(P-value) chromatin tracks representing the epigenetic organization of 111 human tissues from Roadmap Epigenomics Roadmap Epigenomics Consortium et al. (2015) (see Appendix table "predictor_track_descriptions.csv"). These tracks measure the abundance of a particular chromatin mark genome-wide, with smaller (more significant) p-values reflecting a greater abundance of the chromatin mark at a genomic position. Chromatin marks are chemical modifications of histones, the proteins used to package DNA within a cell. We additionally obtained 10 replication timing tracks from the ENCODE consortium. Replication timing assays measure the relative time at which each position in the genome is replicated during cell division. For non-overlapping regions $R$ of predefined size and location (see main text for more details), we extracted the signal for each epigenetic track using 100 bins per region with pybbi Abdennur (2018). We additionally calculated the average nucleotide content in each window by assigning each nucleotide a numeric value between 1 and 5 and taking the average across a bin (N [unspecified nucleotide] = 1, A = 2, C = 3, G = 4, T = 5), and we calculated the GC content as the percent of G and C nucleotides in a bin, resulting in a total of 735 epigenome tracks per region. The mean values for each region were calculated as the mean chromatin signal for each track in the region.

### B.2  MUTATION COUNT DATA

We downloaded somatic single-base substitution mutations identified in the ICGC subset of the Pan-Cancer Analysis of Whole Genomes Consortium cohorts of esophageal adenocarcinoma, skin melanoma, stomach adenocarcinoma, and liver hepatocellular carcinoma. These data are freely available for download from the International Cancer Genomics Consortium data portal (see fig. 4). We excluded mutations on the sex chromosomes (X and Y) because males and females carry different sets of these chromosomes, leading to differential mutation patterns. We summarized the data as mutation counts per window for window sizes of 50bp, 100bp, 500bp, 1kb, 5kb, 10kb, 25kb, 50kb, 100kb, and 1Mb.

### B.3  RESTRICTION TO REGIONS OF HIGH MAPPABILITY

High-throughput genome sequencing works by randomly reading millions of short sequences of nucleotides (36-150 bases in length) from a target genome. These "reads" are then mapped to the human reference genome to reconstruct the target. A challenge is that short sequences of k nucleotides (kmers) can occur multiple times in the genome. This results in ambiguous mappings for some reads and thus a degradation of data quality in regions composed of many kmers that occur multiple times across the genome. Following previous work Polak et al. (2015), we removed regions of the genome with low quality data by calculating a mappability score for each region. Mappability scores reflect how many times a particular kmer occurs in the genome and have been pre-computed for the human reference genome GRCh37. We required that a region's average mappability score based on 36mers

(e.g. average across all sequences of 36 nucleotides in the region) be >70%, reflecting that all 36mers in the region be >70% unique. The majority of the genome passed this threshold; for 10kb regions, for example, >75% of the genome passed this threshold. We chose to measure mappability with 36mers because this was the length of read used to generate the Roadmap Epigenomics sequencing data.

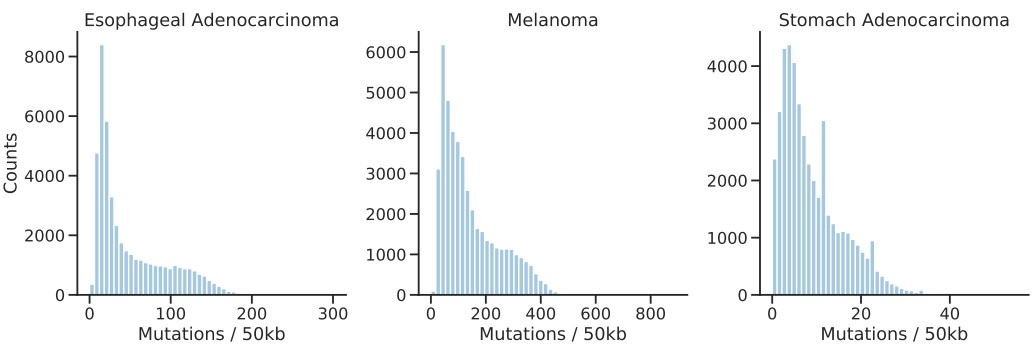

Figure 4: Distribution of mutation counts in 50kb windows tiled across the genome with 36mer uniqueness >70% (see section B.3) for esophageal adenocarcinoma, skin melanoma, and stomach adenocarcinoma. Of note: esophageal adenocarcinoma has a highly skewed distribution, skin melanoma has high mutation counts relative to the other cancers, and stomach adenocarcinoma has low mutation counts relative to the other cancers.

## B.4   SYNTHETIC DATA SIMULATION

We generated synthetic datasets for each of the cancers in order to have datasets with known mean and variance rate parameters. To generate the datasets, we used a k-nearest-neighbors strategy to identify the 500 nearest neighbors for each region. The mean and variance for that region were then taken to be the empirical mean and variance calculated from the 500 nearest neighbors. The number of "observed" mutations was then randomly sampled from a binomial defined by the mean and variance parameters. It is important to note that these datasets are purely derived for the purpose of comparing methods over datasets with a known ground-truth. They do not reflect mutation patterns in the real datasets. The specific steps to generate the simulated data were:

1. Generate vectors of the mean values for each of the 735 tracks (733 epigenetic tracks, GC content track, and average nucleotide content track) in 50kb regions of the genome with 36mer uniqueness >70%.

2. Perform ordinary least-squares (OLS) regression of the mean vectors against the observed number of mutations in each 50kb window for that cancer.

3. Scale each value in the feature vectors by its corresponding coefficient from OLS and compress the weighted mean vectors to 50 components using Principal Components Analysis (capturing >94% of the variance for each cancer).

4. For each region $R$, perform k-nearest-neighbor clustering with Euclidean distance to identify its 500 nearest neighbors in the PC space. Define the mean $\mu_R$ and variance $\sigma_R^2$ of the mutation rate in $R$ to be the mean and variance of the KNN cluster.

5. For region $R$, randomly draw a new "observed" number of mutations from a negative binomial distribution defined using the associated mean and variance. Specifically, $X_R \sim NB(\alpha, 1/(\theta+1))$ where $\alpha = \mu_R^2/\sigma_R^2$ and $\theta = \sigma_R^2/\mu_R$

We created two versions of the simulated data, one in which all regions in the genome were used to estimate the rate parameters and one in which rate parameters were estimated separately within independent train and test subsets. Results were qualitatively indistinguishable.

## C  METHODS

### C.1  GRAPHICAL MODEL

Here we derive the closed form negative binomial distribution presented in the main text as the graphical model marginal distribution over events at some unit $i$ in a region $R$. We use the following notation:

- $M_i$: # mutations observed at pos $i$ (observed)
- $p_i$: genome-wide probability of observing a mutation at the nucleotide context of $i$ (inferred)
- $\tilde{p}_i$: normalized probability of observing a mutation at $i$ in region $R$ (inferred)
- $\lambda_R$: the background mutation rate in region $R$ (unobserved)
- $X_R$: # background mutations in region $R$ (unobserved)
- $\mu_R$: the expected background mutation rate in region $R$ (inferred)
- $\sigma_R^2$: the variance of background mutation rate in region $R$ (inferred).
- $\eta_R$: covariates associated with the behavior of the stochastic process within $R$ (observed)

As presented in the main text and main Figure 1, the graphical model implies the factorization

$$Pr(M_i, X_R, \lambda_R | \alpha_R, \theta_R, \tilde{p}_i; \eta_R) = Pr(M_i = k | X_R, \tilde{p}_i; \eta_R) \cdot Pr(X_R = x | \lambda_R; \eta_R) \cdot Pr(\lambda_R | \alpha_R, \theta_R; \eta_R) \quad (10)$$

where

$$\alpha_R = \mu_R^2 / \sigma_R^2$$
$$\theta_R = \sigma_R^2 / \mu_R.$$

Since $\eta_R$ is a given in each equation, we suppress it for notational ease.

To marginalize out $X_R$, we note that

$$Pr(M_i = k | \lambda_R) = \sum_{x=k}^{\infty} Pr(M_i = k | X_R, \tilde{p}_i) \cdot Pr(X_R = x | \lambda_R)$$

is equivalent to a split Poisson process (Gallager, 2013). Thus

$$Pr(M_i = k | \lambda_R) = \text{Possion}(M_i = k; \tilde{p}_i \lambda_R). \quad (11)$$

We now marginalize out the unknown rate parameter $\lambda_R$.

$$
\begin{aligned}
P(M_i = k | \tilde{p}_i, \alpha_R, \theta_R) &= \int_0^{\infty} P(M_i = k | \lambda_R; \tilde{p}_i) P(\lambda_R | \alpha_R, \theta_R) \mathrm{d}\lambda_R \\
&= \int_0^{\infty} \frac{(\tilde{p}_i \lambda_R)^k}{k!} e^{-\tilde{p}_i \lambda_R} \frac{1}{\Gamma(\alpha_R) \theta_R^{\alpha_R}} \lambda_R^{\alpha_R - 1} e^{-\lambda_R / \theta_R} \mathrm{d}\lambda_R \\
&= \frac{\tilde{p}_i^k}{k! \Gamma(\alpha_R) \theta_R^{\alpha_R}} \int_0^{\infty} \lambda_R^{\alpha_R + k - 1} e^{-\lambda_R(\tilde{p}_i + 1/\theta_R)} \mathrm{d}\lambda_R.
\end{aligned}
$$

Making the substitution $t = \lambda(\tilde{p}_i + 1/\theta_R)$ and noting that the resulting integrand is an unnormalized gamma distribution, we have:

$$
\begin{aligned}
P(M_i = k | \tilde{p}_i, \alpha_R, \theta_R) &= \frac{\tilde{p}_i^k}{k! \Gamma(\alpha_R) \theta_R^{\alpha_R}} \Gamma(\alpha_R + k) \left( \frac{1}{\tilde{p}_i + 1/\theta_R} \right)^{\alpha_R + k} \\
&= \frac{\Gamma(\alpha_R + k)}{k! \Gamma(\alpha_R)} \left( \frac{\tilde{p}_i \theta_R}{\tilde{p}_i \theta_R + 1} \right)^k \left( \frac{1}{\tilde{p}_i \theta_R + 1} \right)^{\alpha_R} \\
&= \text{NB} \left( M_i = k; \alpha_R, \frac{1}{\tilde{p}_i \theta_R + 1} \right).
\end{aligned}
$$

### C.2 OVERVIEW OF PARAMETER ESTIMATION PROCEDURE

*Estimation of regional rate parameters*: As training data, we use a set of input matrices $\{\eta_R; R \in \mathscr{T}\}$ and associated mutation counts $\{X_R; R \in \mathscr{T}\}$. First, a CNN is trained to take $\eta_R$ as input and predict $X_R$ as output, using mean squared error loss. The final 16-dimension feature vector of the trained CNN is then used as input to train a Gaussian process to predict the mutation count $X_R$ and the associated estimation uncertainty by maximizing the likelihood of the observed data. The mean and variance output by the GP were used as estimates for $\mu_R$ and $\sigma_R^2$.

*Estimation of time-averaged event probabilities*: the time-average probability of an event at $p_i$ was estimated based on it's trinucleotide composition, $n, t, n'$ where $n$ is the nucleotide at $i - 1$, $t$ is the nucleotide at $i$ and $n'$ is the nucleotide at $i + 1$ in the reference genome. We first counted every occurrence of $n, t, n'$ in the human genome and then counted the number of times the middle nucleotide of the 3mer was mutated across the genome. The maximum likelihood estimate of $p_i$ is then the ratio of the number of observed mutations of the 3mer divided by the total occurrences of the 3mer.

### C.3 REGIONAL PARAMETERS ESTIMATION METHODS

To compute a model's $R^2$ accuracy to $\mu_R$ and $\sigma_R^2$ for regions $R$ of size $S$, the genome was divided into non-overlapping contiguous segments of size $S$. To assure high data quality, any region with mappability score $< 70\%$ was excluded from further analysis. The remaining windows (accounting for more than 75% of the genome) were randomly divided into train and test sets in an 80–20 split respectively. The test set was held-out and served solely for evaluation purposes. The train set was then divided into train and validation sets by another 80–20 split respectively (train set = 64%, validation = 16%, and test = 20% of the considered regions with mappability score $< 70\%$, see appendix B.3).

#### C.3.1 GAUSSIAN PROCESS FEATURE VECTOR GENERATION

All networks were independently trained for 20 epochs with a batch size of 128 samples and using the Adam optimizer to minimize mean squared error loss to either the true mutation count (CNN and FCNN) or input tensor (AE). After training the model parameters using the train set, predictions over the held-out test set were computed by 1) extracting the last 16-dimensional feature layer (middle feature layer for AE) for all sets over the best performing model over the validation set across all epochs (according to the validation accuracy); 2) training multiple GPs (typically 10) to predict mutation counts using the 16 dimension feature vectors of the train set as input (see appendix C.3.2 for details); 3) taking the mean $\mu_R$ and $\sigma_R^2$ of all 10 runs over the test set as the ensemble prediction of the model. All neural network models were implemented in Pytorch Paszke et al. (2017).

1. **Convolutional neural network (CNN):** The CNN contains 4 convolutional blocks with 2 batch normalized convolutional layers and ReLU activation. The first block transformed the input tesor from $735 \times 100$ to $256 \times 50$ with 256 channels and a double stride. The other blocks are ResNet-style residual blocks that maintain their input dimension to facilitate residual connections, with 256, 512, and 1024 channels respectively. Between each of the 3 residual blocks there is a double stride (ReLU activated and batch normalized) convolutional layer, which divides the tensor length by two and doubles its height with additional channels. The output of the last residual block is flattened and passed through 3 fully-connected layers. The first two are ReLU activated and reduce the dimensionality of the tensor to 128 and 16 dimensions respectively. The last uses linear functions to reduce the tensor to a single cell holding the output of the regression. This forces a linear relation between the regression output and the last feature layer, thus simplifying the function the GP needs to learn, which we found empirically improves the GP's accuracy.

2. **Fully-connected neural network (FCNN):** The FCNN has an architecture similar to the CNN's 3 fully-connected layers but with an input space of the mean epigenetic vector (735 dimensions). Thus, the FCNN is computationally similar to the CNN, but operates on the mean vector instead of the full matrix as an input. The FCNN is designed to demonstrate maximum performance possible when reducing the input tensor to an averaged feature vector.

3. **Autoencoder neural network (AE):** The encoder of the AE used the same architecture as the CNN, excluding the last linear fully connected layer. The decoder has a mirror architecture with the same number of parameters but differs in the internal design of the convolutional blocks. Convolutional layers were replaced by 1D transpose convolutional layers with no batch normalization and no residual connections. The AE was designed to demonstrate the predictive power of a feature embedding that was not optimized to a specific task but produced in a way comparable to the CNN.

4. **Other dimensionality reduction methods:** PCA was computed using the Python Scikit-learn package with default settings and UMAP was computed via Python's umap-learn package McInnes et al. (2018) with 20 nearest neighbours and Euclidean distance. Both methods were computed over the entire training set (80%) with no validation set and reduced the mean epigenetic vector dimensionality (735 dimensions) to 16, just like all other models. Prior to processing, we log-transformed the epigenetic data as we found this improved prediction accuracy downstream.

### C.3.2 GAUSSIAN PROCESS

We implemented a sparse, inducing-point Gaussian process Titsias with a radial basis function kernel using Python's GPyTorch package Gardner et al. (2019). The GP was optimized with 2000 inducing points using the Adam optimizer for 100 steps. All features were mean-centered and standardized to unit variance prior to training. For each dataset, we ran the GP ten independent times and calculated the ensemble mean of the mean and variance predictions from each of the individual runs. We took these ensemble predictions as the mean and variance for each region.

### C.3.3 ALTERNATIVE MODELS

We implemented previously proposed alternative methods Polak et al. (2015); Nik-Zainal et al. (2016); Martincorena et al. (2017) for the estimation of $\mu_R$ and $\sigma_R^2$ without the use of GP. These methods use the mean epigenetic vector as an input.

1. **Random forest (RF):** RF regression was implemented via the Ensemble Methods module in the Python Scikit-learn package, with a maximum depth of 50 trees. Since RF does not directly compute a variance, we implemented the Jackknife method as described in Wager et al. (we have compared our implementation to Polimis et al. (2017) and found them highly correlated). Wager et al. suggests that the number of estimators, i.e., trees, must be linearly related to the number of samples to obtain reasonable estimates of the variance. We chose to have one tenth as many estimators as samples in an attempt to keep running time within reasonable limit for datasets of smaller region sizes. Even so, for 10kb regions (containing approximately 300K regions), RF required >24 hours to train.

2. **Negative binomial regression (NBR):** As described in section 3.3.2 of the main text, NBR directly specifies the variance as $\sigma_R^2 = \mu_R(1 + \beta\mu_R)$, where $\beta$ is an overdispersion parameter. When $\beta = 0$ NBR reduces to Poisson regression, also widely used in the community. NBR was implemented via the discrete module in the Python statsmodels package Seabold & Perktold (2010) with the Broyden–Fletcher–Goldfarb–Shanno optimization algorithm and 1k maximum iterations. Epigenetic predictors were log-transformed and reduced to 20 principle components, following the field-standard Martincorena et al. (2017) in both train and test sets. When used to compare against the GM we also included the expected number of mutations based on the sequence context model (see main paper section 3.2) as an exposure term in the model as in previous work Nik-Zainal et al. (2016); Martincorena et al. (2017).

3. **Binomial regression (BR):** Following a previous study Bertl et al. (2018) that suggested multinomial regression to model multiple types of mutations, we also considered binomial regression (as the binary version of multinomial regression applicable to our simple counts data) as a method to model mutation rates at high resolution. BR was implemented via the generalized linear module in the Python statsmodels package Seabold & Perktold (2010). As in previous work Nik-Zainal et al. (2016); Martincorena et al. (2017), we included the expected number of mutations based on the sequence context model (see main paper section 3.2) as an exposure term in the model. As with NBR, the epigenetic predictors

were log-transformed and reduced to 20 principle components for both train and test sets following state-of-the-art recommendations Martincorena et al. (2017).

## C.4 EMPIRICAL VARIANCE ESTIMATION

For real data, the true variance in mutation counts of a region is unknown. Thus to estimate variance empirically for a given model, we used the following approach:

1. For a region in the test set, perform k-nearest neighbors clustering with Euclidean distance to identify the 500 regions in the train set that are most similar to the region of interest based on the model's feature embedding. For all models, a feature embedding of 16 dimensions was used.

2. Calculate the empirical variance as the variance of the KNN cluster.

Since feature embeddings are model-specific, we calculated an empirical variance estimate per model. The feature-vector embeddings for models specified in section C.3.1 were the feature vectors used as input to the GP. Models specified in section C.3.3 do not create or require comparable feature vectors and therefore were not considered in the main paper results. However, to measure the ability of these methods to estimate empirical variance (Fig. 7), we computed their feature vectors by 1) taking the dot product of the model parameters and the input data mean vectors and 2) reduced these scaled vectors to 16 dimensions via a PCA reduction (explaining 80%-95% of the variance across the different region scales). For RF, we took the model parameters to be the feature importance weights derived from the trained forest and for NBR, we used the model coefficients as the parameters.

## C.5 PERFORMING A GENOME-WIDE SEARCH FOR CANCER DRIVER MUTATIONS

For each cancer, the background mutation rate parameters were estimated across the genome using 5-fold cross validation in 10kb, 25kb and 50kb regions. While the model is robust to choice of 10kb, 25kb or 50kb region size (fig. 5), the 25kb and 50kb models include some additional regions of the genome due to the the mappability threshold (see section B.3). To analyze the largest possible subset of the genome, we performed our analysis iteratively: we first searched for drivers using regions accessible via the 10kb model; we then searched additional regions not accessible by the 10kb model in the 25kb model and then in the 50kb model. To search for drivers, we applied our probabilistic model to estimate the mutation count distributions in 50bp regions across the genome, and we then searched for 50bp regions with significantly more observed mutations than expected under the null distribution of our model. We controlled false-discovery rate at the 0.05 level using a Bonferroni-corrected p-value threshold of P<1e-9.

To compare our hits with known cancer drivers, we tabulated the recurrent driver mutations reported by PCAWG that were present in our dataset, including in the *TERT promoter*, a well known non-coding driver. While most recurrent driver mutations are activating mutations (e.g. cause a gain of cellular function), we also found recurrent mutations in the tumor suppressor genes *TP53* and *SMAD4*. Recurrent mutations in a single position are far less likely in tumor suppressor genes because any deleterious mutation can act as a potential cancer-causing mutation. For example, *TP53* had 6 genome-wide significant 50bp regions, consistent with its status as a crucial tumor suppressor that can be knocked-out with many different mutations (see table 2). Methods specialized to discover driver genes are necessary to find tumor suppressor genes in general Lawrence et al. (2013); Mularoni et al. (2016); Martincorena et al. (2017).

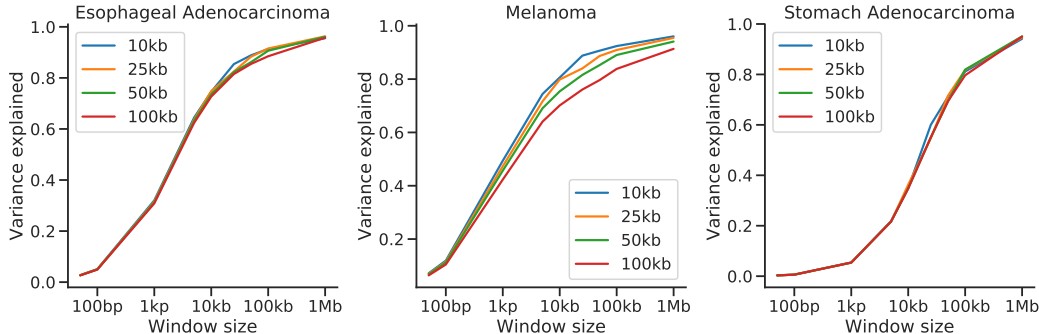

Figure 5: Model robustness to region size. We tested the robustness of our GM estimates to the choice of the scale of region $R$ over which $\mu_R$ and $\sigma_R^2$ were inferred with the CNN+GP. Here we show our GM's Pearson $R^2$ accuracy to the observed number of mutations over a range of sizes for different choices of initial region size $S$. Melanoma shows a slight decrease in performance at larger scales, suggesting local chromatin structure more strongly influences mutation rates in this cancer.

### C.6 ENVIRONMENT AND COMPUTE TIME

A benchmark run at 10kb scale with 10 GP reruns takes 2-3 hours on a single 24 Gb Nvidia RTX GPU, with 8 CPU cores and 756GB RAM. Thus, a full 5-fold of the entire genome takes 10-15 hours. Due to the model's robustness to scale, this time may be significantly reduced without drastic loss of accuracy by using larger region scales (e.g. only 30-40 minutes for 50Kb regions, fig. 5). Importantly, after completing the CNN+GP training, projections to lower or higher scales via the GM require no additional training.

## D APPENDIX RESULTS

### D.1 NEGATIVE BINOMIAL REGRESSION DOES NOT DETECT WELL-KNOWN DRIVERS GENOME-WIDE

Negative binomial regression is the only other method that has been used to perform an unbiased genome-wide search for driver mutations Nik-Zainal et al. (2016); Rheinbay et al. (2020). We thus evaluated how the sensitivity of NBR to detect driver mutations genome-wide compared with the sensitivity of our method. While all known melanoma drivers present in >3 samples were found by the GM by projecting down to only 1kb scale, NBR at 1kb fails to detect *TERT*, the only known common non-coding driver mutation, yielding a p-value that was an order of magnitude less significant than the genome-wide significance for this scale. Similarly, while the GM detects all known esophageal adenocarcinoma drivers by projecting down to 100bp, NBR over 100bp fails to detect *KRAS*, an important genic driver of esophageal cancers, again yielding a p-value that was an order of magnitude less significant than the genome-wide significance threshold for 100bp. Note: we presented results at 50bp in the text to highlight our model's ability to search in arbitrarily small regions, but all known drivers for esophageal adenocarcinoma are also detected in a search over regions of 100bp.

### D.2 CONVOLUTIONAL NEURAL NETWORK OUTPERFORMS OTHER DIMENSIONALITY REDUCTION ALTERNATIVES FOR A GAUSSIAN PROCESS

We first evaluated the methods for regional rate first and second moment inference, $\mu_R$ and $\sigma_R^2$, using our simulated datasets. We calculated accuracy as the Pearson $R^2$ of the estimated mean and variance to the simulated ground-truth mean and variance. CNN+GP, FCNN+GP, NBR and RF accurately inferred $\mu_R$, with $R^2_{\mu_R} > 0.95$ for all three datasets (fig. 6a). However, PCA+GP, UMAP+GP, and AE+GP consistently under-performed (fig. 6a left), suggesting supervision when creating feature vectors is critical for the GP downstream performance.

The CNN+GP and FCNN+GP outperformed the other models when estimating the simulated variance (fig. 6a, right), suggesting the ability to represent arbitrary functions is important for learning

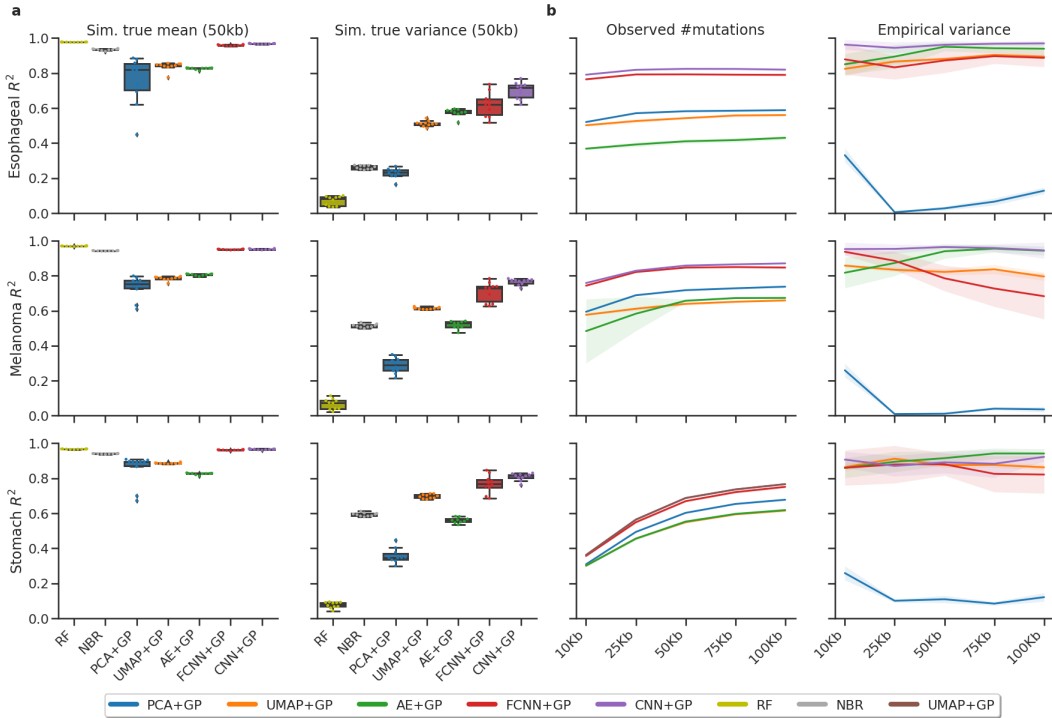

Figure 6: $\mu_R$ and $\sigma_R^2$ estimation accuracy over three cancer types: esophageal adenocarcinoma (top row), skin melanoma (middle row), and stomach adenocarcinoma (bottom row). **a.** $R^2$ accuracy of all models with respect to simulated $\mu_R$ (left) and $\sigma_R^2$ (right) at 50kb. **b.** $R^2$ accuracy of GP-based models to observed number of mutations (left) and empirical variance (right) across scales in real data.

uncertainty in a complex dataset. This conclusion is strengthened by the observation that UMAP and AE enabled relatively accurate variance estimation despite mediocre performance over the mean. Importantly, the clusters used for the simulated data were computed from mean epigenetic vectors; thus our CNN architecture (receiving an input in matrix form) was at a disadvantage. Nonetheless, the CNN+GP most accurately learned both $\mu_R$ and $\sigma_R^2$ across all three simulated datasets (Fig. 6a), with slight improvement over the FCNN+GP.

To further compare the approaches, we applied the GP coupled models to estimate real mutation counts from the three cancers on multiple scales. Models were compared by their $R^2$ to the observed mutations over the test set and to an empirical variance based on the model's own feature vectors (fig. 6b) (see Appendix). The CNN+GP outperformed the FCNN+GP model over observed mutation counts and empirical variance estimation for all three cancer types. Additionally, the performance advantage of the CNN appeared to grow as window size and observed mutation counts increased. This suggests that local epigenetic patterns play an appreciable role in setting mutational processes and indicates that our model is well-designed to leverage the recent growth in genomics corpus sizes.

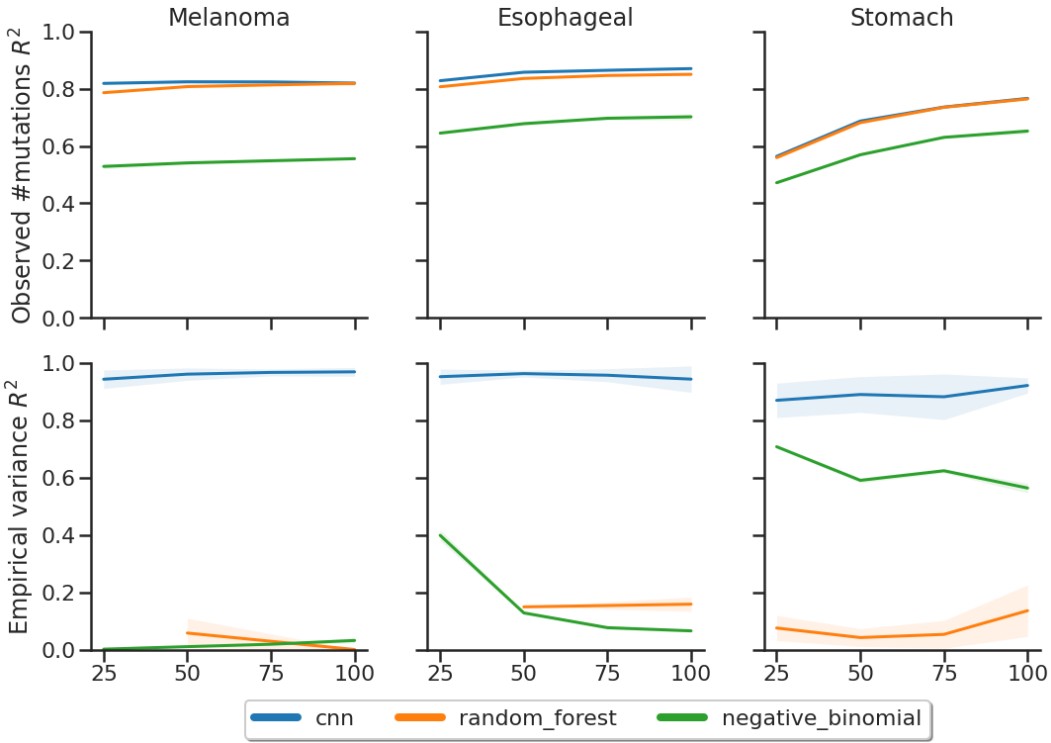

Figure 7: NBR and RF $R^2$ accuracy (with CNN+GP as a reference) to observed number of mutations (top row) and empirical variance (bottom row) in real data and across multiple scales for each cancer type: melanoma (left), esophageal adenocarcinoma (middle) and stomach adenocarcinoma (right). Due to the Jackknife method requirement that the number of RF estimators be linear with respect to the number of samples, estimating RF variance at scale <50kb was computationally infeasible (with >8,000 estimators).

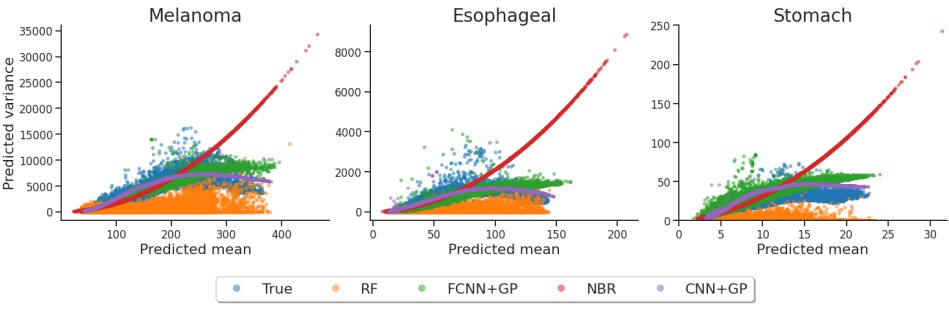

Figure 8: Mean ($\mu_R$) vs variance ($\sigma_R^2$) at 5kb for the ground-truth simulated data (blue) and predictions for each model across all cancer types: melanoma (left), esophageal adenocarcinoma (middle), stomach adenocarcinoma (right). NBR significantly over estimates $\sigma_R^2$ in high mutation count regions because of its strict quadratic relation to the predicted mean. RF consistently underestimates $\sigma_R^2$. FCNN+GP is accurate in low to medium mutation count windows, but overestimates $\sigma_R^2$ with respect to the CNN+GP in high mutation count regions.

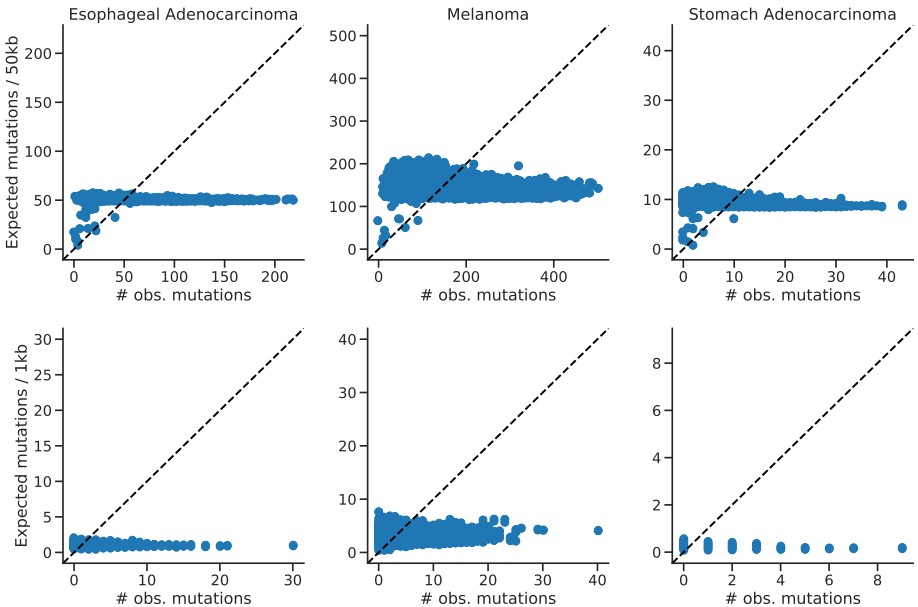

Figure 9: Number of observed mutations versus number of expected mutations based on the sequence context model alone in 50kb and 1kb regions with mappability >70% across the three cancers. Sequence context explains <10% of variance at 50kb and <1% of variance at 1kb scales for all cancers.

## D.3 EXISTING WHOLE-GENOME REGRESSION MODELS ARE TIME INEFFICIENT AT MULTI-RESOLUTION SEARCH

All existing regression models (RF, NBR, BR) require retraining for each desired scale. A requirement that becomes computationally challenging at finer resolutions (e.g. >1.5h for NBR at 100bp). To provide an estimate of the differences between existing methods and our SPG, we performed a multi-scale time analysis presented in . However, it does not include scales <100bp, such as 50bp used in this work to detect driver hot-spots. A log-log transform of the scale against the run-time () exposes a polynomial relation between the the window size and time (for small enough scales where the compute power is not governed by the machine's memory and system operations). Extending this relation to a scale as small as 50bp run-time is as high as  1.5h for BR and  2.5h for NBR. Making the overall run-time for a typical multi-resolution scan of 50bp, 100bp, 500bp, 1kb, 10kb over 2h for BR and over 4h for NBR, while the SPG run-time remains under 1h.

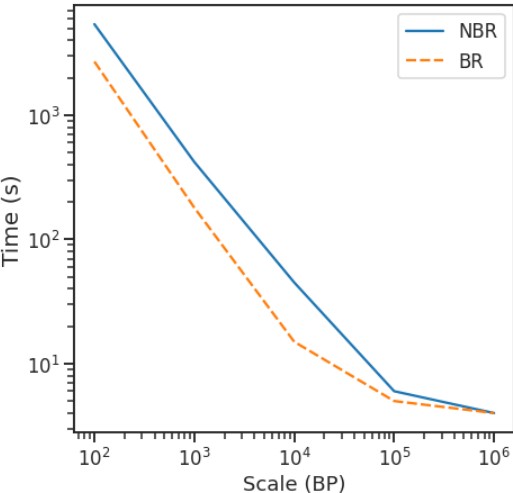

Figure 10: Log-log run-time of current whole-genome regression methods that require retraining per desired scale. Run-time increases polynomially with scale beyond a threshold (where memory operations dominate the computation of the method).

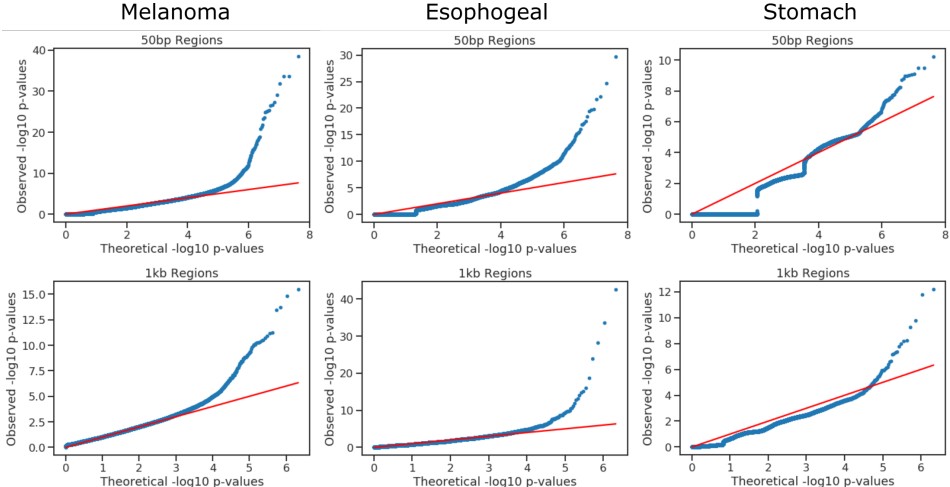

Figure 11: $-log_{10}$(P-value) quantile-quantile (qq) plots for expected vs observed number of mutations in 50bp and 1kb windows using our graphical model with rate parameters estimated in 10kb regions for each cancer. Under a properly calibrated null model, p-values generated from the null distribution are uniformly distributed between zero and one. QQ plots thus provide a qualitative assessment of the accuracy of a model's null distribution: the observed p-values should closely match the expected p-values from a uniform distribution (red line) except at extremely small p-values where observations from the alternate model should be found. The step-like nature of the qq-plot for stomach adenocarcinoma in 50bp regions is because the null distribution is discrete (negative binomial) and the dataset has relatively few mutations; thus each 50bp bin can have only one of a few possible mutation counts (typically between 0 and 5).

| Chrom | Start | End | Observed | Expected | p-value |
|-------|-------|-----|----------|----------|---------|
| 17 | 7577500 | 7577549 | 13 | 0.0141 | $\mathbf{1.75 \times 10^{-30}}$ |
| 17 | 7577100 | 7577149 | 10 | 0.0153 | $\mathbf{7.43 \times 10^{-23}}$ |
| 17 | 7577550 | 7577599 | 8 | 0.00856 | $\mathbf{3.72 \times 10^{-20}}$ |
| 17 | 7578400 | 7578449 | 8 | 0.0147 | $\mathbf{2.76 \times 10^{-18}}$ |
| 17 | 7578500 | 7578549 | 6 | 0.0129 | $\mathbf{6.16 \times 10^{-14}}$ |
| 17 | 7578200 | 7578249 | 6 | 0.0146 | $\mathbf{1.28 \times 10^{-13}}$ |

Table 2: All 50bp windows with significant recurrent mutations in the *TP53* gene from genome wide driver search in esophageal adenocarcinoma.

