# OpenReview forum: "Multi-resolution modeling of a discrete stochastic process identifies causes of cancer"
_ICLR.cc/2021/Conference — ICLR 2021 Poster_

### Official Review · AnonReviewer3 · 2020-10-26
**I vote for weak acceptance as I find that this paper introduces a split-Poisson-Gamma model for discrete stochastic processes at multi-resolutions, that appears to be inremental but may have impacts for the task of the detection of cancer-causaing mutation rates.**

**Rating:** 6
**Confidence:** 1

**Review:**

Short summary: This paper introduces a split-Poisson-Gamma model to capture discrete-time, integer-valued stochastic processes at multiple scales. Although it seems to be simple and incremental compared with Poisson gamma models, this novel method may has some impacts in modeling mutation rates and identifying genomic elements that drive tumor emergence. Thus, I vote for weak acceptance.

-quality: the technique contribution appears to be simple, incremental but could be useful in real applications of detecting cancer-related mutations.

-clarity: the most parts of the paper is clearly written. Nonetheless, I find the paper is improvable by clearly describing the input data in Sec.2.

-originality: to my knowledge, it is the first attempt to develop split-Poisson gamma model for discrete-time, integer-valued stochastic processes at multiple scales.

-significance: it appears to be an incremental contribution in the family of Poisson gamma models. I think the split-Poisson gamma model is a sensible and useful method in detecting cancer-related mutation rates.

Pros: a useful and very sensible model for detecting cancer related mutation rates.

Cons: A clear description of the input data is missing in the beginning of Sec.2. I think it could be better if the authors can position the split-Poisson gamma model among the closely related Poisson gamma models, to clearly distinguish the main differences and why the novel model outperforms the others.

In addition to estimating mu_R and sigma_R^2, how to place hyper priors over these parameters , and perform MAP estimations?

I agree the detection of cancer-causing mutations is a significant application of the multi-resolution modeling of discrete stochastic processes. Nonetheless, I would suggest the authors to discuss the broad applications of the split-Poisson gamma (SPG) model for a venue like ICLR.

comments:
-"NB" notations are used in eq.(6,7) but are not defined for negative binomial yet.
-you miss a "Pr" for lambda_R in Eq.(9).
-can you present the complete procedure  to perform parameter inference in the supplement?
-In C.1 of pp14, it should be NB(M_i;alpha_R, 1/(p_i \theta_R+1)).

---

> ### Author Response · Authors · 2020-11-17
> **Response to reviewer 3 comments**
>
> We thank the reviewer for the close reading of the manuscript and their thoughtful comments. We have revised the paper to address each of the reviewer’s concerns and believe it is substantially improved. Below we describe the changes we have made in response to each concern.
>
> **Comment:** -quality: the technique contribution appears to be simple, incremental but could be useful in real applications of detecting cancer-related mutations.
>
> **Response:** Given the immense size of the human genome and the many hundreds of thousands of statistical tests that often need to be run in application, our modeling choices were heavily motivated by the need to have an accurate linear-time algorithm. We agree that any single piece of the SPG algorithm alone could be considered incremental. However, taken as a whole with CNN+GP (as suggested by Reviewer 2), we believe the approach is an important contribution to the detection of cancer driver mutations as it allows fast, efficient, and accurate searches for drivers anywhere in the genome without the need to retrain a model, which has not previously been possible. We have updated section 1.2 to make this important point clearer.
>
> **Comment:** -clarity: the most parts of the paper is clearly written. Nonetheless, I find the paper is improvable by clearly describing the input data in Sec.2.
>
> **Response:** Thank you for this helpful suggestion. We have added a new paragraph at the start of Section 2, which describes the problem in the abstract, including a description of the input data.
>
> **Comment:** A clear description of the input data is missing in the beginning of Sec.2. I think it could be better if the authors can position the split-Poisson gamma model among the closely related Poisson gamma models, to clearly distinguish the main differences and why the novel model outperforms the others.
>
> **Response:** Section 2 now starts with a description of the input data. In addition, we have added new text to Section 5.2 describing how SPG differs from previous Poisson-Gamma models, particularly negative binomial regression. Section 5.2 now states:
> “The theoretical foundations of negative binomial regression and SPG are similar, both built upon the classical Poisson-gamma model. SPG differs from NBR in three key ways that help explain its improved performance: 1) SPG models mutation patterns over arbitrary sets of positions enabling it to dynamically pool information across positions after a single training; in contrast, NBR operates on fixed regions and must be retrained for every new region size. 2) SPG's variational inference method estimates the gamma parameters for each region independently; NBR estimates only the shape parameter independently for each window and uses a single scale parameter for all windows. 3) SPG's CNN data reduction enables non-linear mapping of spatial covariate information to mutation rate, whereas NBR can perform only linear inference and disregards the spatial organization of the genome.”
>
> **Comment:** In addition to estimating mu_R and sigma_R^2, how to place hyper priors over these parameters , and perform MAP estimations?
>
> **Response:** Since mu_R and sigma_R^2 are inferred using a Gaussian process, placing hyperpriors over these terms is equivalent to the choice of GP kernel function and hyperpriors placed over the kernel parameters. We explored multiple choices of kernel function and hyperpriors and found results were robust to any particular choice. We thus settled on using a radial basis function with no hyperpriors on kernel parameters. We have now added this explanation to section 3.4.
>
> **Comment:** I agree the detection of cancer-causing mutations is a significant application of the multi-resolution modeling of discrete stochastic processes. Nonetheless, I would suggest the authors to discuss the broad applications of the split-Poisson gamma (SPG) model for a venue like ICLR.
>
> **Response:** Thank you for this suggestion. We have now expanded the final paragraph of Discussion to more fully discuss the broader applications of our SPG model. We particularly highlight a possible application to early identification of hotspot outbreaks of COVID-19.
>
> **Comments:** -"NB" notations are used in eq.(6,7) but are not defined for negative binomial yet. -you miss a "Pr" for lambda_R in Eq.(9). -can you present the complete procedure to perform parameter inference in the supplement? -In C.1 of pp14, it should be NB(M_i;alpha_R, 1/(p_i \theta_R+1)).
>
> **Response:** Thank you for pointing out these typos. We have fixed all of them. We have also now included an Appendix section explaining the full parameter inference procedure (C.2).

---

### Official Review · AnonReviewer2 · 2020-10-28
**Well motivated model for specific biological application, with confusing presentation of technical contributions**

**Rating:** 6
**Confidence:** 3

**Review:**

Summary: The paper extends Poisson-Gamma models for non-stationary sequences, in a manner that allows partitioning the counts according to a binomial model to account for multiple resolutions. This generalisation is motivated well with a biological application of practical relevance, and the proposed method is particularly strong in enabling linear computational scaling required for analysis of large genome data.

Reasons for score: I am leaning towards rejection in the current form.  The contribution itself is worth publishing and the method is likely to be valuable for the application, but the presentation would need to be improved (especially regarding the GP+CNN part; see the detailed feedback) to better communicate the technical contributions for the ICLR audience. Conditional on improved presentation, I would be leaning towards acceptance.

Detailed feedback: The proposed split Poisson-Gamma (SPG) process seems reasonable, but in technical terms is a fairly straightforward hierarchical structure and can be constructed using standard properties of Poisson-Gamma. It is well motivated by the resulting efficient inference and this specific application, and may find uses in other applications as well, but does not provide a very clear theoretical contribution that would open immediate follow-up research directions for more general modelling questions.

My main problem with the paper concerns the structure of the presentation. While the authors motivate the model well and provide very clear illustrations for the application, the method sections are disconnected and I had trouble following the technical contributions. In particular, the connection between Section 2 (SPG) and the technical algorithm required for using it (Section 3.3) is unclear. To me it seems the GP+CNN part is an integral part of the overall solution and a contribution in itself (and, in fact, an important one -- the SPG alone is not quite sufficient as theoretical contribution). It provides a concrete algorithm using SPG and is general, but now the description is provided only after talking about specific data and looks more like a minor technical detail with no proper theoretical justification. For me, the paper would be more natural if Section 3.3 (and possibly some other parts of Section 3) would be described after Section 2 as description of how SPG is used in practice, and if they would use shared notation and terminology. This would make the practical approach easier to follow and the contributions more clear.

The empirical experiments and illustrations for the application are well carried out, and serve as good demonstration of the method. However, a reader uninterested or uneducated in this specific application will have some trouble figuring out how well the method works; this could be improved by complementing the results with clear artificial data of slightly simpler nature.

Modifications after discussion:
Increased score by one as the presentation in the revised version has clearly improved, along the lines requested in the original review.

---

> ### Author Response · Authors · 2020-11-17
> **Response to reviewer 2 comments**
>
> We thank the reviewer for carefully reading our manuscript and providing detailed comments on how to improve it. We believe the paper is significantly improved by incorporation of the reviewers’ three suggestions:
> 1. The technical contribution of the work should be more clearly presented to the ICLR audience. In particular, the importance of the CNN+GP should be made clearer.
> 2. To help address Suggestion 1, the methods sections should be reorganized to unify the theoretical derivation of SPG with the technical considerations of how to infer its parameters.
> 3. The inclusion of a clear artificial dataset that is approachable to a reader unfamiliar with the particular application presented in the paper.
>
> Below we describe the steps we have taken to address the suggestions.
>
> 1. We are gratified that the reviewer recognized that achieving efficient inference was of central importance to our downstream application. In practice, cancer biologists often need to test hundreds of thousands of combinations of genomic positions, and our design choices were motivated by the community’s need for efficiency. A crucial contribution of our method is that it enables accurate searches for driver elements and mutations anywhere in the genome without requiring arduous retraining of a model, a feat which is not possible with existing methods. We have revised section 1.2 to make this contribution clearer.
>
>     We also agree with the reviewer that we should have done a better job communicating the theoretical foundations and advances of our inference algorithm to the ICLR audience. The reviewer correctly grasped that the CNN+GP is of central importance to the method, and that our contribution is not the SPG model alone but in fact a unified framework of both a theoretical statistical model and a practical solution to infer its parameters. As suggested by the reviewer, we have reorganized Section 2 in order to present our contributions as a unified whole, describing not only the theory of SPG but also presenting the theoretical foundations for parameter inference using a CNN+GP. Please see the response to Suggestion 2 for further details. We sincerely hope that this improved presentation more clearly communicates our technical contributions.
>
> 2. We have restructured and expanded Section 2 as the reviewer suggested. The new organization is as follows:
>     * Section 2 (intro): abstract definition of the problem that introduces the stochastic process, the prediction task, and notation.
>     * Section 2.1: theoretical foundation of the SPG model (essentially the original section 2).
>     * Section 2.2: the theoretical foundation of regional rate parameter estimation using a GP to perform variational inference. This section also provides background on the use of a CNN prior to training a GP.
>     * Section 2.3: additional information on inferring the long-term time-average event probabilities p_i.
>
>   Section 2 thus presents a full overview of the SPG model, from its theoretical foundations to considerations for parameter estimation independent of application. Section 3 has also been revised to follow the flow of Section 2 and describe a specific example of how parameters can be fit for the application of modeling cancer mutations. We have been careful to unify notations throughout.
>
> 3. The reviewer is correct, it can be onerous to understand this kind of algorithm without a simpler setting. As suggested, we have included a simpler artificial dataset, along with a step-by-step explanation as to how the method works on it.  We have:
>     * Added section 3.1 which explains the artificial dataset in a clearer way.
>     * Added a diagram to Figure 2 that outlines the simulation of and experiments on the artificial dataset.
>     * Added results to Figure 2 that compare the accuracy of estimating the simulated mean between our method and alternative approaches.
>
> We truly hope our improved presentation of both our technical contributions and our artificial dataset makes our work an even better fit for the ICLR audience.

---

> > ### Comment · AnonReviewer2 · 2020-11-23
> > **Main concern addressed by improved presentation**
> >
> > Thank you for the detailed response and for taking my recommendations seriously. The paper now reads better for the ICLR audience, and hence my main concern has been addressed. It is still a paper that requires understanding the application to appreciate the contribution in full and the clarity is still not ideal (I understand in full that the original submitted version naturally constrained you quite a bit while preparing the revision), but the issues are not so severe that they would prevent publication.
> >
> > Overall, I am now leaning towards acceptance, but retain some hesitations due to a bit limited contribution beyond the specific application considered here. While some elements (SPG itself, or the CNN+GP algorithm) may turn out to be useful in a range of tasks, the paper could go further to support that e.g. by better theoretical analysis, more elaborate discussion etc.

---

> > > ### Author Response · Authors · 2020-11-24
> > > **Response to ICLR audiance applicability concern**
> > >
> > > Thank you for your support; we very much appreciate your continued efforts to help us improve the paper. We agree that we can and should make clearer the broad usefulness of the SPG framework and its individual components.
> > >
> > > We have made two changes to the Discussion section to address this:
> > > 1) We have added another Discussion paragraph to describe potential applications of the SPG framework in more detail.
> > > 2) We have added a sentence to the end of Discussion paragraph 1 to highlight the potential broad applicability of the CNN+GP.
> > >
> > > While these are minor changes, we hope it will provide additional clarity to readers.

---

### Official Review · AnonReviewer1 · 2020-10-31
**Split Poisson Gamma distribution to model a discrete stochastic process**

**Rating:** 7
**Confidence:** 3

**Review:**

The authors present the split Poisson Gamma (SPG) distribution, an extension of the Poisson-Gamma distribution, to model a discrete non-stationary stochastic process. SPG has an analytical posterior allowing accurate prediction after the model parameters have been inferred a single time. The authors apply the SPG to model tumor mutation rates and show that model parameters can be accurately inferred from high-dimensional epigenetic data. This is achieved through a combination of CNNs, GPs and MLE. The results are promising in detecting tumor drivers such as genes, regulatory structures and base-pairs.

The paper is well written, with motivation described and prior literature being discussed. Figure 1 is well laid out to drive home the point.

Comments:

* Section 2: are all the distributions univariate? If not, a table giving dimensions will be helpful. What is the dimension of the covariates (eta_R)? It would also help if a plate diagram of the generative model is given.

*  Why was CNN chosen for dimensionality reduction?

*  Were other non neural architectures used for dimensionality reduction?

*  What is the significance of ‘735’ epigenetic tracks?

*  How valid is the assumption that events are distributed independently in the mutation space? I think this is too restrictive but at the same time simplifies the problem. A discussion of this simplification will be essential for the medical/biology audience.

*  Title of paper has 'causes'. Title on ICLR webpage has 'cusses'. Please correct if possible.

---

> ### Author Response · Authors · 2020-11-17
> **Response to reviewer 1 comments**
>
> We thank the reviewer for their careful reading of our manuscript and thoughtful comments. Below we reply to each concern.
>
> **Comment:** Section 2: are all the distributions univariate? If not, a table giving dimensions will be helpful. What is the dimension of the covariates (eta_R)? It would also help if a plate diagram of the generative model is given.
>
> **Response:** Yes, all the distributions are univariate. We have updated the text to specify that all random variables are scalars in the first paragraph of section 2.2.
> In our application the dimension of the covariates is 735x100. We have revised Section 3 to clearly state this dimensionality.
> Thank you for the suggestion to include a plate diagram. We agree this is helpful and have added one as Figure 1e.
>
> **Comment:** Why was CNN chosen for dimensionality reduction?
>
> **Response:** We agree that we did not clearly justify the use of a CNN in the original submission. We have now revised section 3.3, “Estimating dynamic regional rates with uncertainty,” to explain that the columns encode the high-resolution spatial organization of the epigenome which have recently been shown to be important determinants of local mutation rate in particular instances (Gonzalez-Perez et al., 2019; Akdemir et al., 2020). We thus hypothesized that a convolutional neural network would provide a powerful and general approach to produce a low-dimensional embedding that retains information about this local structure; the supervised nature of a CNN further enables the resulting embedding to be optimized for the cancer of interest, which is crucial to performance since the epigenetic organization that determines mutation rate varies drastically between cancer types (Polak et al., 2015). We now clearly present this justification for the use of CNNs in section 3.3.
>
> **Comment:** Were other non neural architectures used for dimensionality reduction?
>
> **Response:** Yes, we investigated multiple other approaches for dimensionality reduction (appendix section C.3.1) including linear regression-based methods, unsupervised methods (PCA and UMAP) and other neural architectures (fully connected and autoencoder). Results from these approaches were included in the original submission as Appendix Figure 6. We have strengthened the language in section 3.3 and section 5.1 to reinforce that we investigated other methods for dimensionality reduction including non-neural architectures and include a pointer to the Appendix.
>
> **Comment:** What is the significance of ‘735’ epigenetic tracks?
>
> **Response:** Thank you for pointing out that this number is mysterious. As we described in Section 3, the ways the genome is chemically modified and organized are crucial determinants of mutation rate. Epigenetic tracks are datasets that assay the presence of these chemical modifications all across the genome. The 735 tracks we use represent the largest set of uniformly processed human epigenome data currently available. We have now added this statement to section 3.1 to clarify to the reader why we chose this particular set of data as input.
>
> **Comment:** How valid is the assumption that events are distributed independently in the mutation space? I think this is too restrictive but at the same time simplifies the problem. A discussion of this simplification will be essential for the medical/biology audience.
>
> **Response:** A good point and one that has been recently considered in the literature: perhaps surprisingly, previous work has established that somatic mutations are approximately independent given the immediate nucleotide composition around a genomic position (Martincorena et al., 2017, Dietlein et al., 2019). Indeed, work from Martincorena and colleagues established that just the trinucleotide composition, as used in our work, is nearly always sufficient to render neighboring positions approximately independent. We have added this information and justification for the independence assumption to the second paragraph of discussion.
>
> **Comment:** Title of paper has 'causes'. Title on ICLR webpage has 'cusses'. Please correct if possible.
>
> **Response:** Thank you for pointing out this embarrassing typo! We have corrected the online title.

---

### Decision · Program_Chairs · 2021-01-07
**Final Decision**

**Decision:**

Accept (Poster)

**Comment:**

Reviewers agree that the paper excels in providing a principle pipeline that combines CNNs and GPs with a Poisson-Gamma distribution to provide a generic approach for multiresolution modelling of tumour mutation rates. As a whole such combination of techniques addresses a key challenge in computational biology that also scales to large datasets.